# Availability and Use of Mobile Health Technology for Disease Diagnosis and Treatment Support by Health Workers in the Ashanti Region of Ghana: A Cross-Sectional Survey

**DOI:** 10.3390/diagnostics11071233

**Published:** 2021-07-09

**Authors:** Ernest Osei, Kwasi Agyei, Boikhutso Tlou, Tivani P. Mashamba-Thompson

**Affiliations:** 1Discipline of Public Health Medicine, School of Nursing and Public Health, University of KwaZulu-Natal, Durban 4001, South Africa; Tlou@ukzn.ac.za (B.T.); Mashamba-Thompson@ukzn.ac.za (T.P.M.-T.); 2School of Business, Spiritan University College, Kumasi, Ghana; a.kwes2@yahoo.com; 3Faculty of Health Sciences, University of Pretoria, Pretoria 0001, South Africa

**Keywords:** mHealth applications, disease diagnosis, treatment support, sub-Saharan Africa

## Abstract

Mobile health (mHealth) technologies have been identified as promising strategies for improving access to healthcare delivery and patient outcomes. However, the extent of availability and use of mHealth among healthcare professionals in Ghana is not known. The study’s main objective was to examine the availability and use of mHealth for disease diagnosis and treatment support by healthcare professionals in the Ashanti Region of Ghana. A cross-sectional survey was carried out among 285 healthcare professionals across 100 primary healthcare clinics in the Ashanti Region with an adopted survey tool. We obtained data on the participants’ background, available health infrastructure, healthcare workforce competency, ownership of a mobile wireless device, usefulness of mHealth, ease of use of mHealth, user satisfaction, and behavioural intention to use mHealth. Descriptive statistics were conducted to characterise healthcare professionals’ demographics and clinical features. Multivariate logistic regression analysis was performed to explore the influence of the demographic factors on the availability and use of mHealth for disease diagnosis and treatment support. STATA version 15 was used to complete all the statistical analyses. Out of the 285 healthcare professionals, 64.91% indicated that mHealth is available to them, while 35.08% have no access to mHealth. Of the 185 healthcare professionals who have access to mHealth, 98.4% are currently using mHealth to support healthcare delivery. Logistic regression model analysis significantly (*p* < 0.05) identified that factors such as the availability of mobile wireless devices, phone calls, text messages, and mobile apps are associated with HIV, TB, medication adherence, clinic appointments, and others. There is a significant association between the availability of mobile wireless devices, text messages, phone calls, mobile apps, and their use for disease diagnosis and treatment compliance from the chi-square test analysis. The findings demonstrate a low level of mHealth use for disease diagnosis and treatment support by healthcare professionals at rural clinics. We encourage policymakers to promote the implementation of mHealth in rural clinics.

## 1. Introduction

Sub-Saharan African (SSA) countries, including Ghana, are confronted with a double burden of communicable and non-communicable diseases [1,2]. They also have weak healthcare systems, which has been exacerbated by the severe acute respiratory syndrome coronavirus 2 (SARS-CoV-2) pandemic [1,3,4,5]. In addition, poor access to healthcare due to insufficient healthcare infrastructure, poor road networks, long-distance travel to health facilities, inadequate health education, lack of financial resources, insufficiently trained health professionals, and many others also further weaken healthcare systems [6,7]. The government of Ghana (GoG) has committed to improving the digitisation of healthcare systems, training and posting many skilled health professionals to rural communities, and expanding mobile networks to rural Ghana [8].

Digitisation of healthcare systems such as mobile health (mHealth) technologies and applications have been identified as promising strategies for improving access to healthcare delivery and patient outcomes [9,10]. Mobile health technology is defined as mobile devices, their various components, and other related technologies in healthcare delivery [11,12]. These applications have been shown to provide a cost-effective, convenient, and broadly accessible modality to implement population-level health interventions [13]. In Ghana, mobile phones’ availability and utilisation as of 2018 was reported to be about 52% and is expected to increase steadily [14]. The high rate of mobile phone penetration and its innovativeness could become a promising tool to enhance healthcare provision and bridge the inequalities of healthcare accessibility [15,16,17]. Mobile phone adoption and acceptability are disproportionately high in resource-limited settings [18]. Thus, mHealth applications can address healthcare disparities among hard-to-reach populations to help achieve universal health for all [19].

Studies in some low- and middle-income countries (LMICs) have indicated that, in this era of SARS-CoV-2, digital health technologies such as mHealth applications have been utilised for screening, diagnosis, risk assessment, tracking of real-time transmissions, and others in all settings [20,21]. The use of mHealth applications could reduce the spread of SARS-CoV-2 and other infectious diseases in overcrowded emergency rooms and improve patient care [19,20,21,22,23]. With the advent of mHealth, patients with chronic diseases could be managed and treated remotely instead of visiting the hospital in-person. Others with acute disease conditions could also be screened and diagnosed remotely with these mHealth applications rather than visiting the overcrowded emergency rooms. This could minimise their risk of contracting SARS-CoV-2 and other infectious diseases in this current condition. Mobile health applications have also been deployed to support disease surveillance, medication, and treatment adherence, improve communication between clinical staff and their patients, appointment reminders, etc., [24,25,26,27,28].

Despite these significant challenges and the limited resources in Ghana, mHealth interventions’ potential in playing a massive transformative role in healthcare provision has received considerable attention [29,30]. Considering the prospects of mHealth applications in resource-limited settings, we conducted a cross-sectional study to determine the availability and use of mHealth applications for disease diagnosis and treatment support by health workers in the Ashanti Region of Ghana. This research focused on the availability of mHealth infrastructure, clinical staff competence, mHealth for diagnostics and treatment, usefulness, ease of use, user satisfaction, and behavioural intention to use mHealth. It is envisaged that the findings of this study will be beneficial to the GoG, donors, non-governmental organisations in health, development partners, and others for improving the quality of healthcare provision by integrating mHealth applications into the normal clinical flow. It is also anticipated that our findings will assist the GoG and other similar settings to implement and sustain digital technologies such as mHealth to promote universal health coverage.

## 2. Methods

### 2.1. Study Design and Participants

A cross-sectional survey was conducted in primary healthcare facilities in the Ashanti Region of Ghana. The researchers conducted this survey to examine the availability and use of mHealth applications for disease diagnosis and treatment support by health professionals in the Ashanti Region of Ghana. In this survey, the participants are healthcare professionals who are highly trained clinical staff such as clinicians, nurses, laboratory scientists, pharmacists, physiotherapists, radiologists, and others mandated to provide healthcare services to the public. Healthcare professionals across 100 health facilities gave written consent to take part in this survey. A few participants were assisted in answering the questionnaire, while the majority answered them independently. All the participants were working in healthcare facilities in the Ashanti Region of Ghana during our survey.

### 2.2. Study Setting

The Ashanti Region is located in the middle part of Ghana (Figure 1). According to the 2010 population census, this region has over 4.70 million inhabitants with a growth rate of 2.7% and is described as Ghana’s business hub [31]. It is projected to reach 9.5 million inhabitants in 2040, according to the Ghana Statistical Service 2012 report [32]. This region is the most populated part of Ghana and has several healthcare facilities that cover the entire region [33]. This region is one area with a high prevalence of several communicable and non-communicable diseases in Ghana. For instance, it has the second-highest prevalence rate of non-communicable diseases such as hypertension, stroke, diabetes, cancer, and others in Ghana [34,35,36,37]. Although this area is the most populated region in Ghana, it is one of the regions with the lowest tuberculosis prevalence rates [38]. Ashanti Region is one of Ghana’s numerous areas with poor healthcare access, especially for people living in poor-resource settings. There are relatively moderate levels of accessibility to general primary healthcare; accessibility to healthcare services remains deficient in several rural districts in this region [39,40]. This is primarily due to the uneven distribution of healthcare facilities since most healthcare facilities are concentrated in urban and semi-urban areas, with few in rural communities [41].

### 2.3. Sampling Method

We obtained a list of 530 primary healthcare facilities from the Ashanti Regional Health Directorate (RHD) of the Ghana Health Service (GHS) [33]. The researchers randomly selected 100 primary healthcare facilities from all 43 districts in the region. Because there are many healthcare facilities across the entire region, 100 healthcare facilities were chosen to ensure comprehensive study coverage. To guarantee the uniformity of sampled primary healthcare facilities in all 43 districts, the following approach was employed: the primary healthcare facilities were first stratified into 43 strata, with each stratum denoting a district in the region. The 530 primary healthcare facilities were grouped into four categories: 167 health centres, 154 clinics, 180 sub-district hospitals, and 29 district hospitals. Probability proportionate to size (PPS) was then used to determine the proportion of healthcare facilities from each stratum and category with the formula: nh = (Nh/N) × n, where nh represents the sample size for each stratum h; Nh represents population size for each stratum h; N represents the total population; and n denotes the total sample size. A purposive sampling technique was used to select all the district hospitals. Based on this, 29 hospitals were selected from Category 1, 30 clinics from Category 2, 28 clinics from Category 3, and 13 clinics from Category 4. We also used proportionate stratification to obtain the total number of primary healthcare facilities selected from the four groups in each of the 43 strata. After that, a simple random sampling technique was employed to select all the 100 healthcare facilities for this study (Appendix A).

### 2.4. Data Collection and Instruments

The researchers adopted the survey tool from studies conducted by Bauer et al. (2014), Bauer et al. (2017), and Abu-Dalbouch (2013) to match our study population, settings, and study aim [42]. The cross-sectional survey tool (Appendix A) was piloted in eight health centres and clinics in the Ashanti Region and modified to suit the local context based on the respondents’ feedback. This pilot study was conducted to ensure the validity, reliability, and precision of data and remove all forms of ambiguity from the survey tool. We collected data on the category of health professionals, type of facility, number of healthcare professionals, number of patients seen per week, available healthcare infrastructure, healthcare workforce competence, ownership of mobile wireless devices, the usefulness of mHealth, ease of use of mHealth, user satisfaction, and behavioural intention to use mHealth. Data were surveyed and collected by the researcher and three trained research assistants.

### 2.5. Ethics Statement

This study was given full ethical clearance from the Biomedical Research Ethics Committee from the University of KwaZulu-Natal (Approval No. BREC/00000202/2019) and Ghana Health Service Ethics Review Committee (Approval No. GHS-ERC006/11/19). Regional clearance and recruitment site clearance of participants were obtained before the data collection commenced. All study participants were given written consent forms explaining the study’s objective, and they signed the informed consent forms prior to their participation.

### 2.6. Outcome Measures

The study focused on examining the availability of mHealth technologies for disease diagnosis and treatment support by health professionals in the Ashanti Region of Ghana. The analysis of this study examined two outcome measures.

The first outcome was the availability of mHealth for disease diagnosis and treatment support which stemmed from the question: “Are there mHealth interventions available in this facility to support healthcare delivery?” A binary response (yes/no) was captured.

The second outcome was the use of mHealth applications for disease diagnosis and treatment support, which stemmed from the question: “What do you use mHealth interventions for?” Responses were captured on four options: find health information, disease diagnosis, treat and manage disease conditions, and treat and monitor patients’ health conditions.

### 2.7. Explanatory Variables

Demographics assessed whether age, sex, categories of health professionals, type of health facility, the total number of healthcare professionals, and the number of patients who visit the facility per week influenced the availability and use of mHealth.Availability of health infrastructure assessed whether health infrastructure availability facilitated the availability of mHealth for diagnostics and treatment support.Healthcare workforce competency assessed whether their level of knowledge influenced the availability and use of mHealth for disease diagnosis and treatment support.Owning a mobile phone or having a mobile phone assessed whether mobile phone ownership facilitated the use of mHealth for diagnostics and treatment support.The usefulness of mHealth assessed whether the benefits of mHealth applications facilitated mHealth for diagnostics and treatment support.Ease of use of mHealth assessed whether the easiness of using mHealth facilitated mHealth for diagnostics and treatment support.User satisfaction of mHealth assessed whether the user satisfaction facilitated mHealth for diagnostics and treatment support.Behavioural intention to use mHealth assessed whether behavioural intention to use mHealth facilitated mHealth for diagnostics and treatment support.

### 2.8. Data Management and Analysis

The completed questionnaires were screened and reviewed by the principal investigator to complete and correct all discrepancies. Data were then captured into a passworded excel spreadsheet. After data cleaning and verification, the data were exported into STATA version 15 which was developed by StataCorp in California, USA. Descriptive statistics such as frequencies, percentages, means, and standard deviations characterise health workers’ demographics and clinical features. They were then presented in tables and others. Multivariate logistic regression was employed to explore the influence of the demographic factors on the availability of mHealth for disease diagnosis and treatment support by healthcare workers. Again, this multivariate logistic regression was also used to explore the influence of the demographic factors on the use of mHealth for disease diagnosis and treatment support by health workers. In the multivariate logistic regression model, a *p*-value of 0.05 was set as the statistical significance. Furthermore, the associations were examined using the odds ratio and 95% CI estimates. A Chi-square test at a significance level of a *p*-value of 0.05 was used to assess the relationship between the availability and the use of mHealth for disease diagnosis and treatment support.

## 3. Results

### 3.1. Characteristics of the Study Participants

This study received a 100% response rate from the healthcare professionals in the selected healthcare facilities in the region. Completed responses were from 285 participants, with 146 males (51.23%) and 139 females (48.77%). The results revealed that the participants aged 31–40 years were the largest share, with 48.07%, followed by those in the category of 20–30 years, with 42.11% of the responses. Participants aged 41–50 and 51–60 years were the smallest shares, with 9.47% and 0.35%, respectively. The largest group (28.7%) of the respondents in this survey were general nurses, while only 2.46% were midwives. Many (49.12%) of the respondents worked at district hospitals, 35.44% worked at health centres at the sub-district level, and 15.44% worked at rural clinics. Mobile health applications are readily available at the district hospitals, followed by the health centres and the rural clinics having poor availability. The average total number of health professionals in each healthcare facility was estimated at 57.8 (95% CI: 20–98). The average number of patients per week seen by these healthcare professionals was 175.4 (95% CI: 74–372) (Appendix A).

### 3.2. Availability of Mobile Health for Diagnostics and Treatment Support in the Ashanti Region

Results from the frequency table (Appendix A) show that mobile wireless devices are available primarily to healthcare professionals with a frequency of 276 (96.84%). Mobile health applications are available with an estimated frequency of 179 (62.81%) and a non-availability frequency of 106 (37.19%). It is also clear that phone calls are the most predominant mHealth technique being utilised by healthcare professionals, with an estimated frequency of 183 (98.92%). Short message service (SMS) is another mHealth intervention used by healthcare professionals with the second highest frequency of 149 (80.54%). Figure 2 illustrates the availability of the various mHealth applications. Again, simple mobile phones are readily available to healthcare professionals with an estimated frequency of 185 (100%), followed by smartphones with 133 (71.89%) and tablets with 107 (57.84%). It is also observed that there is a higher continuous supply of electric power with an estimated frequency of 149 (80.54%) and relatively high available support systems of 106 (57.30%). Furthermore, most healthcare professionals have the requisite skills for diagnostics with a high frequency of 132 (71.36%) and competence for treatment procedures with an estimated frequency of 164 (88.65%).

### 3.3. Use of Mobile Health for Diagnostics and Treatment Support in the Ashanti Region

The frequency table (Appendix A) shows that the high frequency rate of 182 (98.38%) indicates that many healthcare professionals are currently using mHealth applications to promote healthcare delivery. In this region, healthcare professionals use mHealth to support treatment procedures of diseases such as HIV (177, 95.86%), TB (171, 92.43%), hypertension (99, 53.51%), malaria (93, 50.54%), and diabetes (79, 42.70%). Figure 3 demonstrates various diseases that are being treated and managed with mHealth applications. However, only a few healthcare professionals use mHealth to support the treatment of other conditions such as diarrhoea (17, 9.19%), cancer (5, 2.70%), chronic respiratory disease (2, 1.08%), and stroke (0, 0%). In addition, most healthcare professionals use mHealth to search for medical information (117, 63.24%), diagnosis diseases (182, 98.38%), treat and manage disease conditions (162, 87.57%), and treat and monitor patients’ health conditions (144, 77.84%).

Most healthcare professionals agreed that mHealth applications are easy to use when providing healthcare services to their clients. The majority of them confirmed that mHealth applications are easy to use to support disease diagnosis with an estimated frequency of 262 (87.37%). Some other healthcare professionals also indicated that it is flexible to interact with mHealth with an estimated frequency of 273 (95.79%). The survey revealed that healthcare professionals are comfortable using mHealth to support healthcare delivery, with an estimated frequency of 266 (93.33%). In addition, others are very confident in using mHealth with an estimated frequency of 254 (89.12%). Again, some healthcare professionals are delighted with the use of mHealth with an estimated frequency of 218 (76.49%). Moreover, most healthcare professionals would use mHealth to treat and manage patients’ disease conditions with a frequency of 254 (89.12%). Furthermore, others intend to use mHealth for disease diagnosis and treatment support with an estimated frequency of 279 (97.89%).

### 3.4. Availability of Health Infrastructure and Healthcare Workforce Competency

From the multivariate logistic regression model (Table 1), the results illustrate that healthcare workers within the age groups 20–30 (OR = 17.8 (95% CI: 1.49–21.0) and 31–40 (OR = 17.6 (95% CI: 1.45–21.1) had increased odds for toll-free intervention availability when compared to healthcare workers above 40 years. In addition, healthcare workers within the age groups 20–30 and 31–40 had increased odds for mobile apps [OR = 1.46 (95% CI: 0.34–0.18)] and mHealth availability (OR = 2.93 (95% CI: 0.70–12.2) compared to those above 40 years. Male healthcare workers had increased odds for mobile apps’ availability (OR = 1.27 (95% CI: 0.53–3.04), mobile wireless devices (OR = 1.26 (95% CI: 0.11–5.16), and toll-free intervention (OR = 1.02 (95% CI: 0.43–2.41) compared to female healthcare workers (Figure 4). The total number of healthcare professionals with access to available mHealth (OR = 1.16 (95% CI: 1.07–1.25) and mobile apps (OR = 1.09 (95% CI: 1.03–1.17) had increased odds. The results also indicate that the number of patients per week significantly affects mHealth intervention availability, mobile apps, and toll-free intervention. As expected, an increase in the number of patients per week increased the odds of mHealth intervention availability (OR = 1.02 (95% CI: 1.01–1.04) and mobile apps (OR = 1.00 (95% CI: 1.00–1.01) to healthcare workers. However, an increase in patients’ number reduced the odds of toll-free intervention availability (OR = 0.97 (95% CI: 0.96–0.98).

Furthermore, the results show that health professionals such as doctors and pharmacists significantly influenced the requisite skills for diagnostics and competence to use mHealth for treatment support. Doctors had increased odds for the requisite skills for diagnostics (OR = 1.065 (95% CI: 0.45–2.55) and competence to use mHealth for treatment support (OR = 1.153 (95% CI: 0.27–4.88) as compared to laboratory scientists. Pharmacists had increased odds for disease diagnosis requisite skills (OR = 1.243 (95% CI: 0.56–2.71) compared to laboratory scientists. The results also illustrate those district hospitals and health centres significantly affect the supply of power and support systems.

In addition, district hospitals increased the odds for the supply of power (OR = 59.87 (95% CI: 70.06–5117) and support systems (OR =159.7 (95% CI: 4.51–5660) compared to clinics. However, district hospitals had decreased odds (OR = 0.63 (95% CI: 0.11–0.35) for the competence to use mHealth for treatment support. Health centres had increased odds for the supply of power (OR = 53.53 (95% CI: 5.45–525) and support systems (OR =10.68 (95% CI: 1.05–108) compared to clinics. The total number of healthcare professionals with access to smartphones (OR = 1.073 (95% CI: 1.02–1.12) and competence to use mHealth for treatment support (OR = 1.196 (95% CI: 1.09–1.31) had increased odds. However, the total number of healthcare workers with access to power supply (OR = 0.907 (95% CI: 0.85–0.96) and support systems (OR = 0.948 (95% CI: 0.91–0.89) had decreased in odds. Again, an increase in the number of patients per week increased odds for healthcare workers’ competence to use mHealth for treatment support (OR = 1.019 (95% CI: 1.01–1.03) (Appendix A).

### 3.5. Use of mHealth for Diagnostics and Treatment Support

The multivariate model (Table 2) results show that healthcare workers within the age group 20–30 had increased odds for using mHealth to support the treatment of hypertension (OR = 2.28 (95% CI: 0.74–7.05), diabetes (OR = 3.75 (95% CI: 0.96–14.6), cancer (OR = 6.05 (95% CI: 0.01–5.85), and malaria (OR = 1.04 (95% CI: 0.36–3.05) compared to healthcare workers above 40 years. In addition, healthcare workers within the age group 31–40 had increased odds for using mHealth to manage hypertension (OR = 2.12 (95% CI: 0.67–6.68), diabetes (OR = 5.75 (95% CI: 1.43–23.1), cancer (OR = 11.1 (95% CI: 0.01–12.0), and malaria (OR = 1.24 (95% CI: 0.42–3.67) as compared to healthcare workers above 40 years. Being a male healthcare professional raised the odds for the use mHealth to manage HIV (OR = 2.47 (95% CI: 0.37–16.4) and TB (OR = 1.94 (95% CI: 0.49–7.62) compared to being a female healthcare professional. Both medical doctors and nurses had increased odds (OR = 1.66 (95% CI: 0.30–9.16) and (OR = 1.28 (95% CI: 0.28–5.83), respectively for the use of mHealth to manage TB when compared to laboratory scientists (Figure 5).

The results further show that healthcare workers within the age group 20–30 had increased odds for the use of mHealth for disease treatment (OR = 3.05 (95% CI: 0.58–15.9) and using mHealth once a month for diagnostics [OR = 2.16 (95% CI: 0.55–8.55)] and treatment support (OR = 1.06 (95% CI: 0.25–4.43) compared to those above 40 years. In addition, healthcare professionals within the age group 31–40 had a rise in odds for the use of mHealth for disease treatment (OR = 7.25 (95% CI: 1.32–39.9) and using mHealth once a month for diagnostics (OR = 3.64 (95% CI: 0.68–19.3) and treatment support (OR = 2.68 (95% CI: 0.67–10.7). Being a male healthcare worker increased the odds for using mHealth to treat diseases (OR = 1.48 (95% CI: 0.52–4.17), monitor patients’ conditions (OR = 1.22 (95% CI: 0.57–2.59), and mHealth one to six times a week for diagnostics (OR = 1.73 (95% CI: 0.85–3.48) and treatment support (OR = 2.33 (95% CI: 1.03–5.24) compared to being a female healthcare worker.

Medical doctors had decreased odds of using mHealth once a month for treatment support compared to laboratory scientists (OR = 0.38 (95% CI: 0.15–0.19). Again, pharmacists had increased odds for using mHealth application one to six times a week to support treatment (OR = 2.07 (95% CI: 0.98–4.35) compared to laboratory scientists. District hospital increased the odds for the use of mHealth for disease treatment (OR = 1.70 (95% CI: 0.02–13.4) and monitor patients’ conditions (OR = 1.60 (95% CI: 0.05–55.6) compared to clinics. In addition, health centre had increased odds for the use of mHealth for disease treatment (OR = 3.96 (95% CI: 0.23–68.5) and monitoring patients’ conditions (OR = 1.41 (95% CI: 0.20–9.98) when to compared to clinics. As expected, a rise in the number of patients per week increased odds for using mHealth one to six times for diagnostics (OR = 1.01 (95% CI: 0.99–1.01) and treatment support by healthcare workers (OR = 1.01 (95% CI: 1.00–1.01). However, an increase in the number of patients decreased the odds for using mHealth to treat diseases (OR = 0.99 (95% CI: 0.98–0.99) (Appendix A).

### 3.6. Usefulness of mHealth Interventions

The results from the multivariate model (Table 3) suggest that healthcare professionals within the age group 20–30 had reduced odds for the use of mHealth to monitor patients’ disease conditions (OR = 0.15 (95% CI: 0.02–1.07), manage communicable diseases (OR = 0.15 (95% CI: 0.02–1.07), and provide reminders for medication adherence (OR = 0.32 (95% CI: 0.08–1.24) compared to those above 40 years. In addition, healthcare workers within the age group 31–40 had increased odds for the use of mHealth to manage non-communicable diseases (OR = 1.23, (95% CI: 0.54–2.81) and communicable diseases (OR = 1.41(95% CI: 0.54–3.82) as compared to healthcare professionals above 40 years. However, healthcare professionals within the age group 31–40 had reduced odds for the use of mHealth as reminders for the treatment adherence procedures (OR = 0.41(95% CI: 0.17–0.99) when compared to those above 40 years. Male healthcare professionals had increased odds to use mHealth to monitor patients’ disease conditions (OR = 1.76 (95% CI: 0.80–3.85), manage communicable diseases (OR = 1.19 (95% CI: 0.72–2.00), manage non-communicable diseases (OR = 1.19 (95% CI: 0.72–2.10), and as reminders for medication adherence OR = 1.31 (95% CI: 0.67–2.54) when compared to female healthcare professionals.

Medical doctors had three-fold increased odds of using mHealth as reminders for medication adherence compared with laboratory scientists (OR = 3.32 (95% CI: 1.38–7.97). District hospital reduced the odds for the use of mHealth to monitor patients’ disease conditions (OR = 0.41 (95% CI: 0.01–0.78) and as reminders for clinic appointments (OR = 0.18 (95% CI: 0.01–1.02) when compared to clinics. Health centre increased the odds for the use of mHealth to manage communicable diseases as compared to clinics (OR = 1.16 (95% CI: 0.46–2.90). The total number of healthcare professionals who use mHealth as reminders for treatment adherence procedures (OR = 1.04 (95% CI: 1.01–1.08) and clinic appointments (OR = 1.03 (95% CI: 1.00–1.07) had increased odds. A rise in the number of patients per week increased the odds for the use of mHealth to monitor patients’ disease conditions (OR = 1.01 (95% CI: 1.00–1.02) and manage communicable diseases (OR = 1.00 (95% CI: 0.99–1.00).

The results further indicate that healthcare workers within the age group 20–30 had reduced the odds for the use of mHealth for follow-ups (OR = 0.24 (95% CI: 0.07–0.77), test result notifications (OR = 0.35 (95% CI: 0.13–0.95) and making accurate diagnostic decisions (OR = 0.14 (95% CI: 0.01–0.78) compared with those above 40 years. Again, healthcare professionals within the age group 31–40 had increased odds for using mHealth as reminders for drug collection (OR = 1.43 (95% CI: 0.33–6.09) compared with other healthcare workers above 40 years. Male healthcare professionals had increased odds for using mHealth for follow-ups (OR = 1.56 (95% CI: 0.88–2.76) and treating and managing disease conditions (OR = 1.49 (95% CI: 0.83–2.67) when compared to female healthcare professionals. Both medical doctors and nurses had two-fold increased odds of using mHealth to make accurate diagnostic decisions (OR = 2.77 (95% CI: 0.92–8.33), treat and manage disease conditions (OR = 2.67 (95% CI: 1.23–5.77), and increase effectiveness to treat and manage diseases (OR = 2.10 (95% CI: 0.76–5.83) compared with laboratory scientists.

District hospitals increased the odds for mHealth to treat and manage disease conditions than clinics (OR = 1.36 (95% CI: 0.11–16.8). However, as a district hospital, the odds of using mHealth as reminders to collect drugs reduced (OR = 0.24 (95% CI: 0.001–0.28). In addition, a health centre increased the odds for the use of mHealth to notify patients of their test results (OR = 2.39 (95% CI: 0.95–6.03), treat and manage disease conditions (OR = 3.52 (95% CI: 1.28–9.69), and increase the effectiveness for treatment and management of diseases (OR = 3.88 (95% CI: 1.02–14.7) as compared to clinics. The total number of healthcare professionals who use mHealth as reminders for drug collection had increased odds (OR = 1.05 (95% CI: 1.01–1.10). An increase in the number of patients per week increased the odds for the use of mHealth for follow-ups (OR = 1.00 (95% CI: 0.99–1.01) and to increase the effectiveness to treat and manage diseases (OR = 1.00 (95% CI: 1.00–1.01).

### 3.7. Ease of Use of mHealth Interventions

In the multivariate logistic regression model (Table 4), the results demonstrate that healthcare professionals within the age groups 20–30 and 31–40 had increased odds for the flexibility to interact with mHealth devices (OR = 1.16 (95% CI: 0.11–11.8) and easy to use mHealth for treatment support (OR = 1.33 (95% CI: 0.17–10.3) compared to those above 40 years. Being a male healthcare worker increased the odds for mHealth being easy to use for disease diagnosis (OR = 1.71 (95% CI: 0.67–4.29) and having the flexibility to interact with mHealth (OR = 4.00 (95% CI: 0.76–20.9) compared to being a female healthcare professional. Medical doctors had nine-fold increased odds of becoming skilful in using mHealth for disease diagnosis and treatment support (OR = 9.56 (95% CI: 1.78–51.1) compared to laboratory scientists. Again, nurses had two-fold increased odds for mHealth being easy to use for disease diagnosis (OR = 2.66 (95% CI: 0.82–8.62) compared to laboratory scientists.

In addition, district hospital had increased the odds for mHealth being easy to use for disease diagnosis (OR = 14.0 (95% CI: 0.16–11.8) and treatment support (OR = 6.69 (95% CI: 0.02–21.4) compared to clinics. Health centres had increased odds for it being easy to learn how to use mHealth devices (OR = 1.32 (95% CI: 1.79–8.65) and become skilful in using such applications for disease diagnosis and treatment support (OR = 1.32 (95% CI: 0.60–24.3). The total number of healthcare professionals increased the odds for flexibly interacting with mHealth devices for disease diagnosis and treatment support (OR = 1.13 (95% CI: 1.00–1.27). A rise in the number of patients per week increased the odds for easily using mHealth for disease diagnosis (OR = 1.00 (95% CI: 0.99–1.04).

### 3.8. User Satisfaction of mHealth Interventions

The results from the multivariate model (Table 5) show that healthcare workers within the age groups 20–30 and 31–40 had reduced odds for healthcare workers’ confidence in using mHealth for disease diagnosis and treatment support (OR = 0.24 (95% CI: 0.04–1.24) and mHealth increasing the quality of healthcare delivery (OR = 0.18 (95% CI: 0.02–2.07) compared to those above 40 years. Being a male healthcare professional increased the odds of healthcare workers’ comfort (OR = 1.84 (95% CI: 0.65–5.19) and confidence (OR = 2.33 (95% CI: 1.00–5.43) in using mHealth for disease diagnosis and treatment support compared to being female healthcare professional.

Again, medical doctors had increased odds of becoming comfortable using mHealth applications for disease diagnosis and treatment support (OR = 1.02 (95% CI: 0.28–3.80) compared to being a laboratory scientist. Again, nurses had increased odds of feeling comfortable with mHealth (OR = 1.06 (95% CI: 0.29–3.80) and improving the quality of healthcare delivery with mHealth (OR = 2.10 (95% CI: 0.55–7.98) when compared to being laboratory scientists. Health centres had increased odds of healthcare workers’ comfort (OR = 3.84 (95% CI: 0.87–17.8) and confidence (OR = 3.87 (95% CI: 1.13–13.2) with the use of mHealth for disease diagnosis and treatment support compared to clinics. An increase in the number of patients per week increased the odds of using mHealth to improve healthcare delivery quality (OR = 1.01 (95% CI: 0.99–1.02).

### 3.9. Behavioural Intention to Use mHealth Interventions

Results from the multivariate model (Table 5) reveal that healthcare professionals within the age groups 20–30 and 31–40 had increased odds for healthcare professionals intending to use mHealth for the treatment (OR = 0.13 (95% CI: 0.02–0.92) and management of patients’ disease conditions (OR = 0.35 (95% CI: 0.05–2.38) compared to those above 40 years. Being a male healthcare professional increased the odds of healthcare workers’ intention to use mHealth for treating and managing patients’ disease conditions (OR = 2.79 (95% CI: 1.19–6.54) and disease diagnosis and treatment support (OR = 1.97 (95% CI: 1.08–3.60) compared to being female healthcare professional.

In addition, pharmacists had increased odds of healthcare workers’ intention to use mHealth to treat and manage patients’ disease conditions compared to laboratory scientists (OR = 1.45 (95% CI: 0.55–3.82). The odds increased for a district hospital where healthcare workers intend to use mHealth (OR = 2.25 (95% CI: 0.15–32.7) and would always use mHealth for disease diagnosis and treatment support (OR = 3.20 (95% CI: 0.05–24.0) compared to clinics. A rise in the number of patients per week increased the odds for healthcare workers using mHealth to treat and manage patients’ disease conditions (OR = 1.00 (95% CI: 0.99–1.01) and their intention to use mHealth for disease diagnosis and treatment support (OR = 1.01 (95% CI: 0.99–1.04).

### 3.10. Association between Health Infrastructure Availability or Healthcare Workforce Competency and Ownership of Mobile Wireless Devices

A cross-sectional tabulation was done between healthcare infrastructure’s availability or healthcare workforce competency and ownership of mobile wireless devices using a chi-square test (Appendix A). The chi-square test results illustrate a significant relationship between mobile wireless devices’ availability and currently using mHealth to support healthcare provision (*p* < 0.05). Healthcare workers with mobile wireless devices were more likely to use mHealth to support healthcare delivery than those without mobile wireless devices. In addition, the association between mobile wireless devices’ availability and their use to assist malaria conditions’ treatment is statistically significant (*p* < 0.05). Healthcare workers with mobile wireless devices were more likely to use these devices to treat malaria conditions than those without mobile wireless devices.

Moreover, the chi-square test results also show a significant association between mHealth intervention availability and its use to manage malaria conditions (*p* < 0.05). Healthcare professionals with mHealth were more likely to use such interventions to support malaria management than those without mHealth. The results further illustrate a significant relationship between short message services (SMS) and their use to manage hypertension cases (*p* < 0.05). Healthcare workers who stipulated that they have SMS applications were more likely to use such intervention to manage hypertension conditions than those without SMS services. In addition, the chi-square test results suggest a significant relationship between mobile apps and their use to manage TB (*p* < 0.05), diabetes (*p* < 0.05), and disease diagnosis (*p* < 0.05). Healthcare professionals who indicated that they have mobile apps were more likely to use them for diagnosing diseases and managing TB and diabetes conditions than others with no mobile apps. The chi-square test results demonstrate a significant association between toll-free lines and their usage for managing TB (*p* < 0.05) and HIV (*p* < 0.05) conditions. Healthcare workers who suggested that they have toll-free lines were more likely to use this intervention to support the treatment of TB and HIV conditions than others without toll-free lines.

### 3.11. Association between Health Infrastructure Availability or Healthcare Workforce Competency and Usefulness of mHealth Applications

A cross-sectional tabulation was performed between healthcare infrastructure’s availability or healthcare workforce competency and the usefulness of mHealth using a chi-square test (Appendix A). The chi-square test results suggest a significant relationship between mobile wireless devices’ availability and managing non-communicable diseases (NCDs) (*p* < 0.05), communicable diseases (*p* < 0.05), reminders for treatment adherence procedures (*p* < 0.05), clinic appointments (*p* < 0.05), follow-ups (*p* < 0.05), and treating and managing diseases (*p* < 0.05) to support healthcare provision. Healthcare workers with mobile wireless devices were more likely to use these devices to manage communicable and non-communicable diseases, as reminders for treatment adherence procedures, and for clinic appointments and follow-ups than those without mobile wireless devices.

The chi-square test results also show a significant association between the availability of mHealth intervention and its use as reminders for treatment adherence procedures (*p* < 0.05), clinic appointments (*p* < 0.05), follow-ups (*p* < 0.05), and test result notifications (*p* < 0.05) to promote healthcare delivery. Healthcare professionals who indicated that they have mHealth interventions were more likely to use these interventions as reminders for treatment adherence procedures, clinic appointments, follow-ups, and test result notifications than those with no mHealth interventions. The results further illustrate a significant relationship between SMS and their use to manage NCDs (*p* < 0.05) and follow-ups to promote treatment compliance (*p* < 0.05). Healthcare workers with SMS interventions were more likely to use such interventions to manage NCDs and follow-ups than those without mHealth.

### 3.12. Association between Health Infrastructure Availability or Healthcare Workforce Competency and Ease of use of mHealth Applications

A cross-sectional tabulation was done between the availability of healthcare infrastructure or healthcare workforce competency and ease of use of mHealth applications using a chi-square test (Appendix A). The chi-square test results reveal a significant relationship between mobile wireless devices’ availability and the ease of using mHealth for disease diagnosis (*p* < 0.05) and treatment support (*p* < 0.05) and its flexibility (*p* < 0.05) to support healthcare services. Healthcare workers who indicated they have mobile wireless devices were more likely to find it easier and more flexible to use them for disease diagnosis and treatment support than those without mobile wireless devices. In addition, the chi-square test results show a significant association between the availability of mHealth intervention and the ease of using mHealth for treatment support (*p* < 0.05) and its flexibility (*p* < 0.05) to promote healthcare delivery. Healthcare professionals with mHealth were more likely to find it easier and flexible to use these mHealth interventions to support patients’ disease diagnosis and treatment conditions than others with no mHealth.

The results further show a significant relationship between SMS and its ease of using mHealth for treatment support (*p* < 0.05) and its flexibility (*p* < 0.05) to enhance the provision of quality healthcare. Healthcare professionals with SMS interventions were more likely to find it easier and flexible to use such interventions for disease diagnosis and treatment support than those without mHealth. In addition, the chi-square test results show a significant relationship between phone calls and its ease of using mHealth for disease diagnosis (*p* < 0.05) and treatment support (*p* < 0.05), its flexibility (*p* < 0.05), becoming skilful in using mHealth (*p* < 0.05), and it being easy to learn how to use mHealth (*p* < 0.05). Healthcare workers who indicated that they use phone/voice call interventions were more likely to find it easier and flexible to use such applications for disease diagnosis and treatment support than healthcare workers without access to voice calls.

### 3.13. Association between Health Infrastructure Availability or Healthcare Workforce Competency and User Satisfaction of mHealth

A cross-sectional tabulation was done between healthcare infrastructure’s availability or healthcare workforce competency and user satisfaction of mHealth using a chi-square test (Appendix A). The chi-square test results show a significant relationship between mobile wireless devices’ availability and confidence (*p* < 0.05) and being completely satisfied (*p* < 0.05) using mHealth for disease diagnosis and treatment support. Healthcare professionals who were confident and completely satisfied with mHealth were more likely to use these mobile wireless devices for disease diagnosis and treatment support than those with no confidence in mHealth. In addition, the chi-square test results illustrate a significant association between the availability of mHealth intervention and comfort with mHealth (*p* < 0.05), confidence in mHealth (*p* < 0.05), and increase quality healthcare (*p* < 0.05). Healthcare professionals with mHealth interventions who were comfortable and confident in mHealth were more likely to use such applications to boost quality healthcare delivery than others with no mHealth interventions.

The results further show a significant relationship between SMS and completely satisfied with mHealth applications (*p* < 0.05). Healthcare workers with SMS applications were more likely to be happy with mHealth for disease diagnosis and treatment support than others with no SMS application access. In addition, the chi-square test results indicate a significant association between phone calls and their comfortability (*p* < 0.05) and increase quality healthcare (*p* < 0.05). Healthcare professionals who suggested using phone call interventions were more likely to be comfortable using mHealth to boost quality healthcare delivery than those without access to voice calls. Again, the chi-square test results illustrate a significant relationship between mobile apps and completely satisfied using mHealth applications (*p* < 0.05). Healthcare workers with mobile apps were more likely to be happy with mHealth for disease diagnosis and treatment support than others with no mobile apps. The chi-square test results reveal a significant association between toll-free intervention and comfort using mHealth (*p* < 0.05). Healthcare professionals who have access to toll-free lines were more likely to be comfortable using mHealth than those without toll-free lines.

### 3.14. Association between Health Infrastructure Availability or Healthcare Workforce Competency and Behavioural Intention to use mHealth

A cross-sectional tabulation was performed between the availability of healthcare infrastructure or healthcare workforce competency and behavioural intention to use mHealth using a chi-square test (Appendix A). The chi-square test results illustrate a significant relationship between mobile wireless devices’ availability and always using mHealth for disease diagnosis and treatment support (*p* < 0.05). Healthcare professionals with mobile wireless devices were more likely to use mHealth for disease diagnosis and treatment support than others with no mobile wireless devices. Additionally, the chi-square test found a significant association between mHealth intervention availability and always using mHealth for disease diagnosis and treatment support (*p* < 0.05). Healthcare workers with mHealth interventions were more likely to use mHealth for disease diagnosis than others with no mHealth interventions.

Furthermore, the results also show a significant relationship between SMS and the ability to use mHealth to treat and manage patients’ conditions (*p* < 0.05). Healthcare professionals with SMS interventions were more likely to use mHealth to treat and manage patients’ needs than others with no SMS intervention access. The chi-square test found a significant association between phone calls and intention to use mHealth for disease diagnosis and treatment support (*p* < 0.05). Healthcare workers who use phone call interventions intended to use mHealth for disease diagnosis and treatment support than those with no access to voice calls. Again, the chi-square test results demonstrate a significant relationship between mobile apps and always use mHealth for disease diagnosis and treatment support (*p* < 0.05). Healthcare professionals with mobile apps were more likely to use mHealth for disease diagnosis and treatment support than those without mobile apps.

## 4. Discussion

This study aimed to examine the availability and use of mHealth applications for disease diagnosis and treatment support by healthcare workers in Ghana. In this study, 64.91% of healthcare professionals indicated that mHealth applications are available to them, while 35.08% do not have access to mHealth. In addition, 98.38% of healthcare professionals are currently using mHealth applications to support healthcare delivery. The findings show that mobile wireless devices such as simple mobile phones, smartphones, and tablets are readily available to healthcare professionals in Ghana’s Ashanti Region. The results also reveal that mHealth applications such as phone or voice calls, SMS, mobile apps, and toll-free lines are available to healthcare workers and are currently being used to support healthcare delivery. The results further illustrate that healthcare professionals predominantly use mHealth applications to screen or diagnose many existing disease conditions in this region.

Additionally, the results demonstrate that healthcare workers in this part of Ghana currently use mHealth to treat HIV, TB, hypertension, diabetes, malaria, and diarrhoea conditions. Again, the results reveal that healthcare professionals continuously use mHealth to support healthcare provision due to the constant supply of power. Moreover, the findings suggest that most healthcare professionals have the requisite skills and competence in using mHealth applications for diagnostics and treatment procedures of disease conditions. Furthermore, the results demonstrate a low-level use of mHealth applications for disease diagnosis and treatment support by healthcare professionals at rural clinics.

A study conducted in the USA largely agrees with this current survey where healthcare workers use mHealth to treat and manage chronic diseases such as HIV, TB, hypertension, and diabetes, among others [42]. This current survey results fully support the findings from similar surveys conducted in primary care clinics in the USA [43]. In their studies, most healthcare workers were comfortable and confident in using mHealth applications to support their patients’ healthcare needs [43]. The findings demonstrate that healthcare workers use mHealth applications to promote medication adherence, clinic appointments, and follow-ups. This corroborates with the findings from a similar study conducted by Belcher et al. in Saudi Arabia, where mHealth applications improved the treatment of diabetes, clinic appointments, and check-ups [44].

This current study’s limitations include that respondents’ inclusion was based on patient consent, which may have introduced selection bias into the study sample. Due to the limited funding for the data collection, only 285 participants were enrolled from this region’s numerous primary healthcare clinics. Our current results may not be generalised beyond the Ashanti Region of Ghana among healthcare professionals using mHealth for disease diagnosis and treatment support. Despite all these limitations, our current study is, to the best of our knowledge, the first comprehensive research on the availability and use of mHealth for disease diagnosis and treatment support by healthcare professionals in the Ashanti Region of Ghana. The study helped determine the current availability and use of mHealth applications by healthcare professionals to diagnose and treat diseases in this region. This could guide policymakers in formulating guidelines on the utilisation of mHealth technologies to promote quality healthcare delivery.

This current study achieved its primary objective and demonstrated a gap in mHealth for disease diagnosis and treatment support at rural clinics in Ghana’s Ashanti Region. This means that policymakers and implementors should adopt various strategies to facilitate the implementation of mHealth applications for disease diagnosis and treatment support in such resource-constrained settings and enhance their scale-ups. Given this, we recommend a proposed framework for improving the implementation of mHealth for disease diagnosis and treatment support in low- and middle-income countries (LMICs) [12]. The results show that mHealth applications are generally available to healthcare professionals and are being utilised for disease diagnosis and treatment support of patients’ conditions. This is a good sign that the continuous use of mHealth should be strengthened to promote quality healthcare delivery as recommended by the World Health Organisation (WHO) 2019 guidelines on digital health [45].

The results demonstrate a low-level use of mHealth applications for disease diagnosis and treatment support by healthcare professionals at rural clinics. To this end, we encourage policymakers to deliberately implement mHealth at rural clinics to support disease diagnosis and treatment procedures of patients’ conditions. The findings show that healthcare professionals employed mHealth to treat diseases such as HIV, TB, hypertension, and diabetes. We recommend that more primary studies be conducted focused on using mHealth to treat and manage other diseases such as cancer, stroke, chronic respiratory conditions, asthma, and others in this region. Moreover, the findings indicate that most healthcare professionals use mHealth applications to screen or diagnose several common disease conditions in this region. Hence, we encourage healthcare professionals to use mHealth interventions to screen or diagnose several other neglected tropical diseases to enhance early detection to initiate proper treatment processes.

## 5. Conclusions

The study revealed that mHealth applications are primarily available to healthcare professionals to promote quality healthcare delivery in the Ashanti Region. The findings show that healthcare professionals use mHealth applications to screen or diagnose, treat, and manage several common disease conditions at primary healthcare clinics. The results also demonstrate a low-level use of mHealth applications for disease diagnosis and treatment support by healthcare professionals at rural clinics. Future studies are recommended to examine the availability and use of mHealth applications for disease diagnosis and treatment support by healthcare professionals at rural clinics.

## Figures and Tables

**Figure 1 diagnostics-11-01233-f001:**
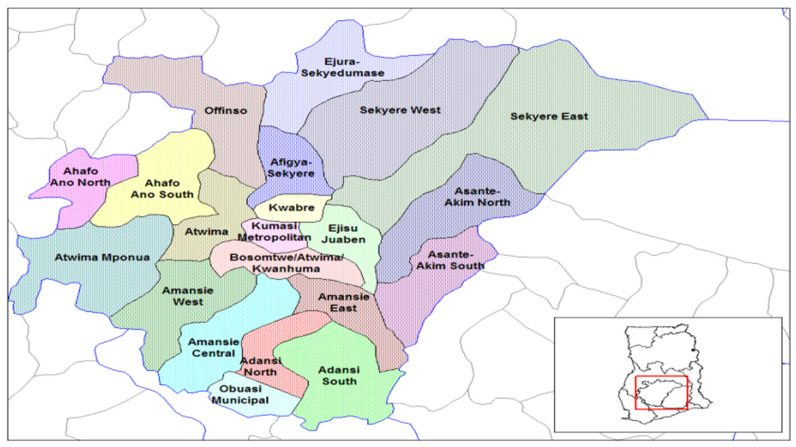
Map of Ashanti Region of Ghana.

**Figure 2 diagnostics-11-01233-f002:**
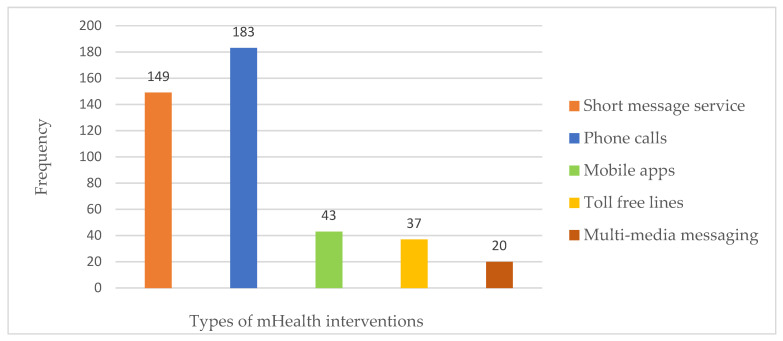
Availability of the various mHealth applications.

**Figure 3 diagnostics-11-01233-f003:**
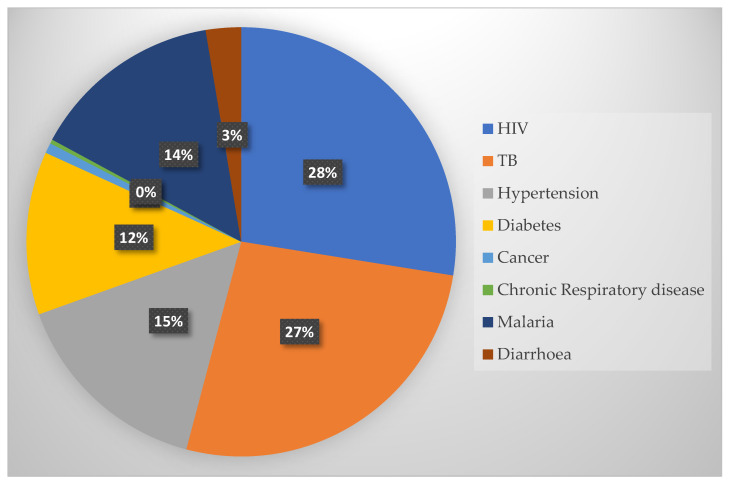
Types of diseases treated and managed with mHealth applications.

**Figure 4 diagnostics-11-01233-f004:**
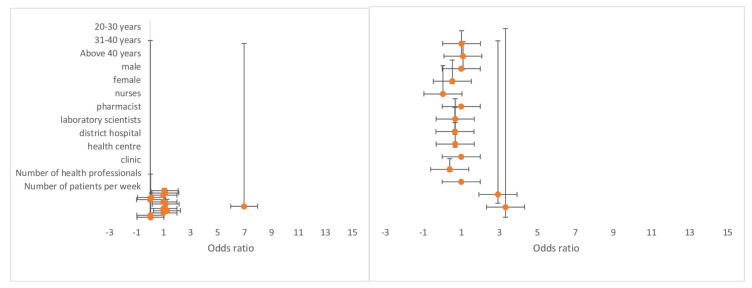
Odds ratio showing the association on the availability of mobile wireless devices and mHealth applications for disease diagnosis and treatment support by health workers in Ashanti region, Ghana.

**Figure 5 diagnostics-11-01233-f005:**
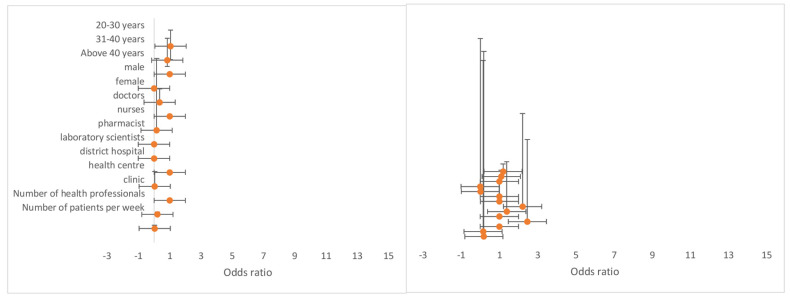
Odds ratio showing the association on the use of mHealth applications for the management and treatment of HIV and TB conditions by health workers in Ashanti region, Ghana.

**Table 1 diagnostics-11-01233-t001:** Multivariate analysis results for the availability of health infrastructure and healthcare workforce competency.

**Variable**		**Mobile Wireless Devices**				**mHealth Availability**				**SMS**			**Mobile Apps**				**Toll-Free**		
		**Odds Ratio**	**95% CI**		***p*-Value**	**Odds Ratio**		**95% CI**	***p*-Value**	**Odds Ratio**	**95% CI**	***p*-Value**	**Odds Ratio**	**95% CI**	***p*-Value**		**Odds Ratio**	**95% CI**	***p*-Value**
Age	20–30 years	0.040	0.01–19.6		0.733	3.33		0.82–14.2	0.104	0.66	0.13–3.31	0.622	1.46 **	0.34–6.18	0.010		17.8 **	1.49–21.0	0.02
	31–40 years	0.020	0.03–78.8		0.648	2.93 **		0.70–12.2	0.010	1.49	0.29–7.68	0.632	2.21	0.51–9.64	0.287		17.6 **	1.45–21.1	0.02
	Above 40 years	1				1				1			1				1		
Sex	Male	1.260 **	0.11–5.16		0.02	0.40		0.18–0.85	0.018	0.81	0.34–1.96	0.653	1.27 **	0.53–3.04	0.010		1.02 **	0.43–2.41	0.002
	Female	1				1				1			1				1		
Categories of health professionals	Doctors	-				0.68		0.23–2.03	0.500	1.00	0.34–2.95	0.992	1.16	0.40–3.31	0.781		1.48	0.53–4.15	0.44
	Nurses	6.980	0.66–73.5		0.106	0.67		0.23–1.91	0.458	1.52	0.48–4.85	0.471	0.84	0.27–2.55	0.764		0.96	0.33–2.83	0.95
	Pharmacists	1.160	0.27–6.06		0.843	0.68		0.28–1.61	0.385	1.27	0.39–4.15	0.688	0.83	0.31–2.22	0.720		0.84	0.35–2.02	0.71
	Laboratory Scientists	1				1				1			1				1		
Type of healthcare facility	District hospital	0.010	0.45–1.28		0.993	0.04		0.01–2.26	0.119	2.16	0.04–1.00	0.693	-	-	-		23.8	0	0.98
	Health centre	0.087	0.32–2.01		0.996	0.53		0.17–1.68	0.287	0.59	0.08–4.13	0.600	-	-	-		84.7	0	0.98
	Clinic	1				1				1		1	1				1		
Total number of healthcare professionals		1.140	0.98–1.35		0.089	1.09 **		1.03–1.17	0.004	1.01	0.97–1.07	0.470	1.16 ***	1.07–1.25	<0.001		1.01	0.95–1.07	0.60
Total number of patients per week		1.070	1.02–1.13		0.004	1.02 ***		1.01–1.04	<0.001	0.99	0.99–1.00	0.885	1.00 **	1.00–1.01	0.020		0.97 ***	0.96–0.98	<0.001
**Variable**		**Smartphones**			**Tablets**			**Supply of Power**			**Support Systems**			**Requisite Skills**			**Competence to Use mHealth**		
		**Odds Ratio**	**95% CI**	***p*-Value**	**Odds Ratio**	**95% CI**	***p*-Value**	**Odds Ratio**	**95% CI**	***p*-Value**	**Odds Ratio**	**95% CI**	***p*-Value**	**Odds Ratio**	**95% CI**	***p*-Value**	**Odds Ratio**	**95% CI**	***p*-Value**
Age	20–30 years	0.729	0.20–2.64	0.631	1.418	0.44–4.54	0.556	0.978	0.24–3.96	0.976	0.654	0.21–2.05	0.468	0.298	0.07–1.21	0.090	0.595	0.63–5.61	0.651
	31–40 years	0.823	0.22–3.05	0.771	1.769	0.54–5.83	0.348	1.491	0.35–6.26	0.585	0.871	0.27–2.78	0.817	0.604	0.14–2.47	0.485	1.805	0.16–20.1	0.631
	Above 40 years	1			1			1			1			1			1		
Sex	Male	0.917	0.44–1.89	0.814	0.985	0.49–1.97	0.968	1.075	0.45–2.55	0.870	1.209	0.63–2.31	0.565	1.208	0.59–2.46	0.601	0.671	0.20–2.24	0.516
	Female	1			1			1			1			1			1		
Categories of health professionals	Doctors	1.132	0.47–2.71	0.779	1.242	0.54–2.87	0.612	0.386	0.12–1.15	0.090	1.143	0.52–2.52	0.740	1.065 **	0.45–2.55	0.003	1.153 **	0.27–4.88	0.004
	Nurses	1.236	0.49–3.08	0.649	1.690	0.45–2.56	0.881	0.345	0.11–1.06	0.065	0.787	0.35–1.77	0.563	0.723	0.30–1.73	0.467	0.345	0.08–1.39	0.136
	Pharmacists	0.924	0.39–2.17	0.857	1.280	0.56–2.94	0.561	0.487	0.18–1.29	0.149	0.792	0.36–1.72	0557	1.243**	0.56–2.71	0.010	0.654	0.26–1.61	0.358
	Laboratory Scientists	1			1			1			1			1			1		
Type of healthcare facility	District hospital	0.872	0	0.984	0.119	0.008–2.94	0.193	59.87 ***	70.06–5117	<0.001	159.7 **	4.51–5660	0.005	21.66	0.73–639	0.075	0.623 ***	0.11–0.35	<0.001
	Health centre	0.1333	0	0.987	0.409	0.59–2.81	0.364	53.53 ***	5.45–525	0.001	10.68 **	1.05–108	0.045	2.777	0.43–17.8	0.282	0.630	0.04–9.74	0.741
	Clinic	1			1			1			1			1			1		
Total number of healthcare professionals		1.073 **	1.02–1.12	0.002	1.057	1.02–1.10	0.007	0.907 ***	0.85–0.96	0.001	0.948 **	0.81–0.89	0.015	0.969	0.93–1.01	0.160	1.196 ***	1.09–1.31	<0.001
Total number of patients per week		0.997	0.99–1.00	0.413	1.000	0.99–1.001	0.953	0.997	0.99–1.00	0.418	1.002	0.99–1.00	0.315	0.998	0.99–1.00	0.447	1.019 ***	1.01–1.03	0.001

**: *p*-value < 0.05: *** *p*-value < 0.001. Source: Author’s computation based on data obtained from the field survey, 2020.

**Table 2 diagnostics-11-01233-t002:** Multivariate analysis results for the use of mHealth for diagnostics and treatment support.

**Variable**		**Ever Used or Currently Using mHealth**			**HIV**			**TB**			**Hypertension**			**Diabetes**			**Cancer**			**Malaria**		
		**Odds Ratio**	**95% CI**	***p*-Value**	**Odds Ratio**	**95% CI**	***p*-Value**	**Odds Ratio**	**95% CI**	***p*-Value**	**Odds Ratio**	**95% CI**	***p*-Value**	**Odds Ratio**	**95% CI**	***p*-Value**	**Odds Ratio**	**95% CI**	***p*-Value**	**Odds Ratio**	**95% CI**	***p*-Value**
Age	20–30 years	0.04	0.05–0.22	0.866	0.19	0.01–37.0	0.535	0.16	0.003–8.91	0.37	2.28 **	0.74–7.05	0.011	3.75 **	0.96–14.6	0.054	6.05 **	0.01–5.85	0.006	1.04 **	0.36–3.05	0.010
	31–40 years	0.02	0.08–0.16	0.737	0.15	0.001–34.2	0.496	0.26	0.004–14.5	0.506	2.12 **	0.67–6.68	0.020	5.75 **	1.43–23.1	0.014	11.1 **	0.01–12.0	0.004	1.24 **	0.42–3.67	0.002
	Above 40 years	1			1			1			1			1			1			1		
Sex	Male	0.05	0.12–1.05	0.527	2.47 **	0.37–16.4	0.003	1.94 **	0.49–7.62	0.034	0.84	0.44–1.69	0.60	1.37	0.71–2.63	0.350	0.54	0.07–4.14	0.550	0.96	0.51–1.79	0.893
	Female	1			1			1			1			1			1			1		
Categories of health professionals	Doctors	-	0	0	1.39	0.19–9.98	0.739	1.66 **	0.30–9.16	0.054	0.78	0.36–1.69	0.527	0.81	0.37–1.79	0.609	0.85	0.09–7.37	0.884	0.99	0.47–2.12	0.998
	Nurses	-	0	0	2.22	0.27–18.6	0.459	1.28 **	0.28–5.83	0.046	0.53	0.23–1.19	0.124	0.57	0.25–1.31	0.187	0.14	0.01–2.15	0.157	0.61	0.27–1.34	0.215
	Pharmacists	0.17	0.005–5.13	0.309	1.01	0.26–3.91	0.981	1.13	0.41–3.15	0.806	1.19	0.56–2.52	0.648	1.26	0.61–2.60	0.528	0.85	0.14–5.04	0.859	1.41	0.69–2.88	0.352
	Laboratory Scientists	1			1			1			1			1			1			1		
Type of healthcare facility	District hospital	0.36	0.13–0.96	0.509	0.01	0.29–30.6	0.259	18.1 **	0.05–6.94	0.003	2.47 **	0.11–57.0	0.053	0.12	0.04–3.35	0.212	-	0	0	0.36	0.02–7.85	0.514
	Health centre	-	0	0	0	0	0	3.89	0.42–36.6	0.233	1.06	1.16–7.01	0.947	0.45	0.64–3.15	0.421	-	0	0	0.33	0.05–2.14	0.244
	Clinic	1			1			1			1			1			1			1		
Total number of healthcare professionals		0.85	0.45–1.56	0.596	1.10 **	0.97–1.25	0.010	0.98	0.89–1.07	0.615	1.00	0.96–1.04	0.843	1.04 **	0.99–1.09	0.053	0.96	0.83–1.10	0.556	1.00	0.97–1.04	0.842
Total number of patients per week		1.06	0.96–1.16	0.228	1.2 **	1.00–1.50	0.019	1.2 **	1.00–74.0	0.012	0.99	0.99–1.00	0.147	0.99	0.99–1.00	0.351	0.97 **	0.94–0.99	0.011	0.99	0.99–1.00	0.710
**Variable**		**Medical Information**			**Disease Treatment**			**Monitor Patients’ Conditions**			**Once a Month for Diagnostics**			**1 to 6 Times a Week for Diagnostics**			**Once a Month for Treatment Support**			**1 to 6 Times a Week for Treatment Support**		
		**Odds Ratio**	**95% CI**	***p*-Value**	**Odds Ratio**	**95% CI**	***p*-Value**	**Odds Ratio**	**95% CI**	***p*-Value**	**Odds Ratio**	**95% CI**	***p*-Value**	**Odds Ratio**	**95% CI**	***p*-Value**	**Odds Ratio**	**95% CI**	***p*-Value**	**Odds Ratio**	**95% CI**	***p*-Value**
Age	20–30 years	0.67	0.21–2.31	0.500	1.06	0.25–4.43	0.939	0.64	0.16–2.45	0.513	3.05 **	0.58–15.9	0.018	0.74	0.24–2.31	0.602	2.16 **	0.55–8.55	0.002	0.71	0.20–2.49	0.593
	31–40 years	0.79	0.25–2.57	0.706	7.25 **	1.32–39.9	0.023	1.04	0.26–4.16	0.959	3.64 **	0.68–19.3	0.012	0.92	0.28–2.92	0.888	2.68 **	0.67–10.7	0.013	0.91	0.25–3.26	0.886
	Above 40 years	1			1			1			1			1			1			1		
Sex	Male	1.32	0.69–2.53	0.397	1.48 **	0.52–4.17	0.041	1.22 **	0.57–2.59	0.002	1.17	0.56–2.46	0.662	1.73 **	0.85–3.48	0.012	0.66	0.33–1.32	0.245	2.33 **	1.03–5.24	0.040
	Female	1			1			1			1			1			1			1		
Categories of health professionals	Doctors	1.19	0.54–2.54	0.669	0.93	0.26–3.28	0.913	1.35	0.54–3.37	0.742	0.56	0.22–1.40	0.214	0.87	0.38–2.01	0.757	0.38 **	0.15–0.91	0.030	1.24	0.49–3.13	0.641
	Nurses	1.11	0.48–2.54	0.810	0.45	0.13–1.63	0.226	1.17	0.46–2.92	0.753	1.48	0.61–3.59	0.385	1.00	0.42–2.38	0.996	0.79	0.34–1.84	0.592	1.39	0.52–3.76	0.506
	Pharmacists	2.29	0.90–5.83	0.081	1.32	0.43–4.02	0.623	0.89	0.42–1.86	0.792	0.64	0.26–1.62	0.352	1.69	0.83–3.44	0.143	0.69	0.31–1.56	0.385	2.07 **	0.98–4.38	0.054
	Laboratory Scientists	1			1			1			1			1			1			1		
Type of healthcare facility	District hospital	0.35	0.01–10.8	0.552	1.70 **	0.02–13.4	0.011	1.60 **	0.05–55.6	0.028	6.43	0.18–234	0.310	0.28	0.01–7.88	0.461	0.88	0.03–26.3	0.944	0.76	0.02–27.3	0.885
	Health centre	0.42	0.80–1.50	0.457	3.96 **	0.23–68.5	0.003	1.41 **	0.20–9.98	0.010	1.34	0.19–9.29	0.761	0.54	0.08–3.74	0.532	1.50	0.22–10.2	0.679	0.39	0.05–2.98	0.370
	Clinic	1			1			1		1	1			1			1			1		
Total number of healthcare professionals		0.99	0.50–3.42	0.858	1.00	0.95–1.06	0.905	1.00	0.96–1.05	0.487	0.98	0.93–1.02	0.309	1.01	0.96–1.05	0.659	1.01	0.96–1.05	0.747	0.98	0.94–1.03	0.508
Total number of patients per week		1.00	0.98–3.48	0.941	0.99 **	0.98–0.99	0.026	1.00	0.99–1.00	0.839	0.99	0.98–1.00	0.099	1.01**	0.99–1.01	0.054	0.99	0.99–1.00	0.288	1.01**	1.00–1.01	0.020

**: *p*-value < 0.05: *** *p*-value < 0.001. Source: Author’s computation based on data obtained from the field survey, 2020.

**Table 3 diagnostics-11-01233-t003:** Multivariate analysis results for the usefulness of mHealth interventions.

**Variable**		**Monitor Patients’ Disease Conditions**			**Manage Non-Communicable Diseases**			**Manage Communicable Diseases**			**Reminders for Treatment Adherence Procedures**			**Reminders for Medication Adherence**			**Reminders for Clinic Appointments**		
		**Odds Ratio**	**95% CI**	***p*-Value**	**Odds Ratio**	**95% CI**	***p*-Value**	**Odds Ratio**	**95% CI**	***p*-Value**	**Odds Ratio**	**95% CI**	***p*-Value**	**Odds Ratio**	**95% CI**	***p*-Value**	**Odds Ratio**	**95% CI**	***p*-Value**
Age	20–30 years	0.15 **	0.02–1.07	0.052	0.91	0.39–2.11	0.831	0.91 **	0.39–2.11	0.034	0.52	0.21–1.29	0.160	0.32 **	0.08–1.24	0.011	0.46	0.13–1.58	0.219
	31–40 years	0.24	0.04–1.58	0.140	1.23 **	0.54–2.81	0.011	1.41 **	0.54–3.82	0.301	0.41 **	0.17–0.99	0.049	0.35	0.09–1.37	0.132	0.40	0.12–1.36	0.143
	Above 40 years	1			1			1			1			1			1	0.67–2.41	
Sex	Male	1.76 **	0.80–3.85	0.015	1.19 **	0.72–2.00	0.004	1.19 **	0.72–2.10	0.026	1.05	0.62–1.76	0.840	1.31 **	0.67–2.54	0.023	1.27	0.68–2.41	0.446
	Female	1			1			1						1			1		
Categories of health professionals	Doctors	2.42	0.84–6.94	0.099	1.37	0.67–2.78	0.378	1.37	0.67–2.78	0.124	0.84	0.41–1.76	0.638	3.32 **	1.38–7.97	0.007	1.22	0.51–2.95	0.644
	Nurses	2.42	0.51–3.53	0.545	1.42	0.70–2.86	0.328	1.42	0.70–2.86	0.388	0.60	0.30–1.22	0.165	2.02	0.89–4.59	0.090	0.55	0.24–1.28	0.168
	Pharmacists	1.34	0.56–3.19	0.507	1.28	0.65–2.50	0.471	1.28	0.65–2.50	0.323	0.77	0.41–1.45	0.421	0.85	0.43–1.68	0.645	0.51	0.24–1.09	0.086
	Laboratory Scientists	1			1			1			1			1			1		
Type of healthcare facility	District hospital	0.41 **	0.01–0.78	0.035	2.46	0.26–22.7	0.427	2.46	0.26–22.7	0.777	0.22	0.02–2.28	0.206	0.15	0.01–4.71	0.119	0.18 **	0.01–1.02	0.051
	Health centre	1.15	0.28–4.63	0.836	1.16	0.4–-2.90	0.750	1.16 **	0.46–2.90	0.036	1.09	0.43–2.79	0.844	1.16	0.28–4.71	0.829	1.89	0.63–5.66	0.253
	Clinic	1			1			1		1	1			1			1		
Total number of healthcare professionals		1.07	0.98–1.08	0.134	1.16	0.95–1.02	0.571	0.99	0.95–1.02	0.667	1.04 **	1.01–1.08	0.017	1.01	0.97–1.05	0.427	1.03 **	1.00–1.07	0.046
Total number of patients per week		1.01 **	1.00–1.02	0.003	0.99	0.99–1.00	0.141	1.00 **	0.99–1.00	0.020	0.99	0.99–1.00	0.894	1.00	0.99–1.00	0.382	1.00	0.99–1.01	0.443
**Variable**		**Reminders for Drugs Collection**			**Follow-ups**			**Test Result Notification**			**Treating and Managing disease Conditions**			**Accurate Diagnostic Decisions**			**Increase Effectiveness for Treatment and Management of Diseases**		
		**Odds Ratio**	**95% CI**	***p*-Value**	**Odds Ratio**	**95% CI**	***p*-Value**	**Odds Ratio**	**95% CI**	***p*-Value**	**Odds Ratio**	**95% CI**	***p*-Value**	**Odds Ratio**	**95% CI**	***p*-Value**	**Odds Ratio**	**95% CI**	***p*-Value**
Age	20–30 years	0.36	0.09–1.42	0.148	0.24 **	0.07–0.77	0.017	0.35 **	0.13–0.95	0.039	0.73	0.27–1.98	0.545	0.14 **	0.01–0.78	0.030	0.36	0.08–1.65	0.191
	31–40 years	1.43 **	0.33–6.09	0.024	0.36	0.11–1.18	0.093	0.42	0.16–1.11	0.083	0.62	0.23–1.66	0.345	0.15	0.01–0.90	0.041	0.41	0.09–1.91	0.263
	Above 40 years	1			1			1			1			1			1		
Sex	Male	1.34	0.62–2.90	0.446	1.56 **	0.88–2.76	0.012	1.35	0.80–2.28	0.249	1.49 **	0.83–2.67	0.016	1.23	0.55–2.74	0.600	1.43	0.67–3.05	0.346
	Female	1			1			1			1			1			1		
Categories of health professionals	Doctors	0.98	0.35–2.73	0.977	0.67	0.31–1.46	0.321	1.19	0.59–2.42	0.611	1.68	0.80–3.53	0.164	2.77 **	0.92–8.33	0.004	1.16	0.44–3.02	0.754
	Nurses	0.48	0.18–1.29	0.148	0.84	0.38–1.84	0.672	0.99	0.49–1.98	0.979	2.67**	1.23–5.77	0.012	1.45	0.55–3.83	0.443	2.10 **	0.76–5.83	0.015
	Pharmacists	0.69	0.32–1.51	0.363	0.65	0.31–1.38	0.269	0.94	0.50–1.77	0.865	0.74	0.39–1.42	0.380	0.52	0.24–1.16	0.112	1.62	0.58–4.51	0.348
	Laboratory Scientists	1			1			1			1			1					
Type of healthcare facility	District hospital	0.24 **	0.001–0.28	0.004	0.57	0.06–5.65	0.636	0.55	0.07–4.30	0.571	1.36 **	0.11–16.8	0.010	0.30	0.02–5.51	0.421	0.73	0.03–17.9	0.848
	Health centre	1.62	0.38–6.86	0.514	2.39	0.87–6.56	0.089	2.39 **	0.95–6.03	0.054	3.52 **	1.28–9.69	0.015	0.72	0.11–4.57	0.732	3.88 **	1.02–14.7	0.046
	Clinic	1			1			1			1			1			1		
Total number of healthcare professionals		1.05 **	1.01–1.10	0.011	1.00	0.96–1.03	0.880	1.01	0.98–1.03	0.482	1.00	0.96–1.03	0.975	1.00	0.95–1.04	0.925	0.99	0.94–1.04	0.873
Total number of patients per week		1.00	0.99–1.01	0.270	1.00 **	0.99–1.01	0.053	1.00	0.99–5.39	0.153	1.00	0.99–1.00	0.459	1.00	0.99–1.00	0.521	1.00**	1.00–1.01	0.024

**: *p*-value < 0.05: *** *p*-value < 0.001. Source: Author’s computation based on data obtained from the field survey, 2020.

**Table 4 diagnostics-11-01233-t004:** Multivariate analysis results for the ease of use of mHealth, user satisfaction of mHealth, and behavioural intention to use mHealth interventions.

Variable		Easy to Use mHealth for Disease Diagnosis			Easy to Use mHealth for Treatment			Flexible to Interact with mHealth			Frustrating to Interact with mHealth			Easy to Become Skilful in Using mHealth			Easy to Learn how to use mHealth Devices for Diagnosis and Treatment		
		Odds Ratio	95% CI	*p*-Value	Odds Ratio	95% CI	*p*-Value	Odds Ratio	95% CI	*p*-Value	Odds Ratio	95% CI	*p*-Value	Odds Ratio	95% CI	*p*-Value	Odds Ratio	95% CI	*p*-Value
Age	20–30 years	0.15	0.02–1.15	0.069	0.69	0.09–4.96	0.715	1.16 **	0.11–11.8	0.010	1.16	0.11–11.8	0.895	0.66	0.11–3.85	0.647	0.23	0.03–1.96	0.182
	31–40 years	0.40	0.05–3.09	0.386	1.33 **	0.17–10.3	0.027	0.52	0.04–7.12	0.627	0.52	0.038–7.12	0.627	0.83	0.14–5.13	0.847	0.26	0.03–2.20	0.218
	Above 40 years	1			1			1			1			1			1		
Sex	Male	1.71 **	0.67–4.29	0.025	0.91	0.27–3.05	0.883	4.00 **	0.76–20.9	0.010	0.40	0.76–20.9	0.100	1.94	0.68–5.51	0.213	1.72	0.59–4.91	0.315
	Female	1			1			1			1			1			1		
Categories of health professionals	Doctors	2.81	0.83–9.48	0.095	2.27	0.36–14.3	0.381	-	0	0	-	0	0	9.56 **	1.78–51.1	0.008	2.12	0.57–7.79	0.257
	Nurses	2.66 **	0.82–8.62	0.010	0.62	0.13–2.78	0.534	2.25	0.30–16.7	0.427	2.25	0.62–5.51	0.427	3.20	0.91–11.3	0.070	2.41	0.65–8.90	0.186
	Pharmacists	1.71	0.52–5.61	0.370	0.92	0.31–2.63	0.868	1.85	0.62–5.51	0.268	1.85	0.16–2.10	0.268	1.57	0.45–5.47	0.477	0.89	0.35–2.25	0.810
	Laboratory Scientists	1			1			1			1			1			1		
Type of healthcare facility	District hospital	14.0 **	0.16–11.8	0.020	6.69 **	0.02–21.4	0.051	0.57	0.16–2.10	0.075	0.58	0.02–4.20	0.075	3.78	0.01–45.5	0.586	1.40	0.08–22.3	0.310
	Health centre	2.29	0.40–12.9	0.347	1.61	0.16–16.0	0.683	0.31	0.02–4.20	0.376	0.30	1.00–1.27	0.376	3.83 **	0.60–24.3	0.015	1.32 **	1.79–8.65	0.011
	Clinic	1			1			1		1	1			1			1		
Total number of healthcare professionals		0.95	0.89–1.02	0.185	0.96	0.88–1.05	0.421	1.13 **	1.00–1.27	0.041	1.15	1.00–1.29	0.430	0.98	0.91–1.05	0.653	0.95	0.89–1.04	0.361
Total number of patients per week		1.00 **	0.99–1.04	0.021	1.00	0.99–1.01	0.740	0.98	0.96–1.00	0.111	0.98	0.96–1.00	0.111	1.00	0.99–1.01	0.573	1.00	0.99–1.01	0.363

**: *p*-value < 0.05: *** *p*-value < 0.001. Source: Author’s computation based on data obtained from the field survey, 2020.

**Table 5 diagnostics-11-01233-t005:** Multivariate Analysis Results for the User Satisfaction of mHealth Interventions.

**Variable**		**Comfortable in Using mHealth for Disease Diagnosis and Treatment Support**			**Confident in Using mHealth for Disease Diagnosis and Treatment Support**			**Completely Satisfied with Using mHealth for Disease Diagnosis and Treatment Support**			**Using mHealth Increases the Quality of Healthcare Delivery**		
		**Odds Ratio**	**95% CI**	***p*-Value**	**Odds Ratio**	**95% CI**	***p*-Value**	**Odds Ratio**	**95% CI**	***p*-Value**	**Odds Ratio**	**95% CI**	***p*-Value**
Age	20–30 years	0.13	0.01–1.32	0.085	0.24 **	0.04–1.24	0.002	0.81	0.31–2.13	0.60	0.24	0.02–2.76	0.253
	31–40 years	0.33	0.03–3.42	0.358	0.60	0.11–3.17	0.552	0.89	0.34–2.29	0.813	0.18 **	0.02–2.07	0.012
	Above 40 years	1			1			1			1		
Sex	Male	1.84 **	0.65–5.19	0.024	2.33 **	1.00–5.43	0.049	1.48	0.84–2.63	0.177	1.33	0.47–3.73	0.584
	Female	1			1			1			1		
Categories of health professionals	Doctors	1.02 **	0.28–3.67	0.010	0.44	0.14–1.36	0.155	0.88	0.39–1.96	0.767	1.60	0.42–6.08	0.484
	Nurses	1.06 **	0.29–3.80	0.021	0.43	0.14–1.34	0.146	0.80	0.36–1.75	0.584	2.10 **	0.55–7.98	0.024
	Pharmacists	0.76	0.29–1.95	0.573	1.02	0.43–2.45	0.959	0.99	0.49–2.00	0.993	1.12	0.35–3.57	0.847
	Laboratory Scientists	1			1			1			1		
Type of healthcare facility	District hospital	0.38	0.82–17.8	0.619	0.36	0.02–7.59	0.514	0.78	0.08–7.46	0.834	0.40	0.01–15.3	0.626
	Health centre	3.84 **	0.87–17.8	0.006	3.87 **	1.13–13.2	0.031	1.68	0.65–4.30	0.277	2.40	0.46–12.3	0.295
	Clinic	1			1			1		1	1		
Total number of healthcare professionals		1.03	0.97–1.09	0.236	1.02	0.98–1.07	0.262	1.01	0.97–1.04	0.524	1.01	0.94–1.06	0.797
Total number of patients per week		0.99	0.99–1.00	0.818	1.00	0.99–1.01	0.380	1.00	0.99–1.01	0.135	1.01 **	0.99–1.02	0.053
**Multivariate Analysis Results for the Behavioural Intention to Use mHealth Interventions**													
**Variable**		**Use mHealth for the Treatment and Management of Patients’ Disease Conditions**					**Always Using mHealth for Disease Diagnosis and Treatment Support**				**Intend to Use mHealth for Disease Diagnosis and Treatment Support**		
		**Odds Ratio**	**95% CI**	***p*-Value**			**Odds Ratio**	**95% CI**	***p*-Value**		**Odds Ratio**	**95% CI**	***p*-Value**
Age	20–30 years	0.13 **	0.02–0.92	0.041			0.83	0.31–2.17	0.705		0.16	0.06–13.4	0.349
	31–40 years	0.35 **	0.05–2.38	0.024			1.21	0.46–3.14	0.694		0.39	0.02–58.9	0.718
	Above 40 years	1					1				1		
Sex	Male	2.79 **	1.19–6.54	0.018			1.97 **	1.08–3.60	0.026		2.47	0.35–17.2	0.359
	Female	1					1				1		
Categories of health professionals	Doctors	0.95	0.31–2.87	0.933			1.23	0.54–2.77	0.617		0.70	0.08–5.95	0.751
	Nurses	0.59	0.21–1.69	0.328			0.97	0.44–2.14	0.947		0.98	0.10–9.26	0.990
	Pharmacists	1.45 **	0.55–3.82	0.044			1.04	0.52–2.12	0.896		1.10	0.35–3.41	0.860
	Laboratory Scientists	1					1				1		
Type of healthcare facility	District hospital	0.52	0.02–12.9	0.691			2.25 **	0.15–32.7	0.052		3.70 **	0.05–24.0	0.007
	Health centre	1.08	0.22–5.16	0.923			0.81	0.25–2.60	0.728		-	0	0
	Clinic	1					1				1		1
Total number of healthcare professionals		0.99	0.95–1.04	0.968			0.98	0.94–1.02	0.493		0.96	0.85–1.10	0.622
Total number of patients per week		1.00 **	0.99–1.01	0.0201			1.00	0.99–1.00	0.892		1.01 **	0.99–1.04	0.006

**: *p*-value < 0.05: *** *p*-value < 0.001. Source: Author’s computation based on data obtained from the field survey, 2020.

## Data Availability

Data for this study are the property of the University of KwaZulu-Natal and can be made available publicly. All interested persons can access the dataset from the author Ernest Osei via this email address (ernestosei56@gmail.com) and the University of KwaZulu-Natal Biomedical Research Ethics Committee (BREC) using the following contacts: The Chairperson Biomedical Research Ethics Administration Research Office, Westville Campus, Govan Mbeki Building University of KwaZulu-Natal P/Bag X54001, Durban, 4000 KwaZulu-Natal, South Africa (Tel.: +27-31260-4769; Fax: +27-31260-4609; Email: BREC@ukzn.ac.za).

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
