# Peer review of "Availability and Use of Mobile Health Technology for Disease Diagnosis and Treatment Support by Health Workers in the Ashanti Region of Ghana: A Cross-Sectional Survey"

_diagnostics, 2021, doi:10.3390/diagnostics11071233_

Round 1

Reviewer 1 Report

The study aims to examine the availability and use of mHealth for disease diagnosis and treatment support by healthcare professionals in the Ashanti Region of Ghana. The research design and methods are sound. The results are well described however, the authors should work on the figures as they are blurry.

The authors should proofread the manuscript for minor spell checks.

The authors should reference the manuscript according to the journals referencing style.

Author Response

Authors’ response to reviewer’s comments

Journal: Diagnostics

Manuscript ID: diagnostics-1244483

Title: Availability and Use of Mobile health technology for disease diagnosis and treatment support by health workers in the Ashanti region of Ghana: A cross-sectional survey

Dear Reviewer

Thank you for reviewing our manuscript titled: Availability and Use of Mobile health technology for disease diagnosis and treatment support by health workers in the Ashanti region of Ghana: A cross-sectional survey and providing constructive feedback.  We have considered your comments and recommendations. Please, find below our responses to your comments, queries, and suggestions. All revisions have been highlighted in yellow in our main manuscript.  Any further comments will be addressed and modified by us.

Reviewer 1

General comments: The study aims to examine the availability and use of mHealth for disease diagnosis and treatment support by healthcare professionals in the Ashanti Region of Ghana. The research design and methods are sound.

Comment 1: The results are well described however, the authors should work on the figures as they are blurry.

Response: The fonts of the figures have been amended. Kindly refer to pages 5, 10, 12, 17 & 24

Comment 2: The authors should proofread the manuscript for minor spell checks.

Response: All the minor grammatical errors have been amended accordingly.

Comment 3: The authors should reference the manuscript according to the journals referencing style

Response: The reference has been formatted according to the journal’s referencing style.

Reviewer 2 Report

In this manuscript, the authors conducted a cross-sectional survey and investigated the availability and use of mobile health technology for the purpose of disease diagnosis and treatment support by health care professionals in Ghana. The study was carefully conducted and the manuscript generally well written. I do not have particular criticism on the paper except a few minor suggestions.

1), Abstract, the reporting of the logistic regression results can be more informative by specifying what factors are associated with the outcome variables. Similarly, the results of the chi-square test can also be clarified. The authors did not mention anything about "rural primary healthcare clinics" in the results part, however, they suddenly raised this issue in the discussion part, which is weird and should be revised.

2), Introduction: third paragraph, can the authors give more explanation on why the use of mHealth applications can reduce the spread of SARS-CoV-2 and other infectious diseases?

3), Figures 2-5 are hard to read, the fonts are too small.

4), there are many formatting problems with the references.

Author Response

Authors’ response to reviewer’s comments

Journal: Diagnostics

Manuscript ID: diagnostics-1244483

Title: Availability and Use of Mobile health technology for disease diagnosis and treatment support by health workers in the Ashanti region of Ghana: A cross-sectional survey

Dear Reviewer 

Thank you for reviewing our manuscript titled: Availability and Use of Mobile health technology for disease diagnosis and treatment support by health workers in the Ashanti region of Ghana: A cross-sectional survey and providing constructive feedback.  We have considered your comments and recommendations. Please, find below our responses to your comments, queries, and suggestions. All revisions have been highlighted in yellow in our main manuscript.  Any further comments will be addressed and modified by us.

Reviewer 2

General comments: In this manuscript, the authors conducted a cross-sectional survey and investigated the availability and use of mobile health technology for the purpose of disease diagnosis and treatment support by health care professionals in Ghana. The study was carefully conducted and the manuscript generally well written. I do not have particular criticism on the paper except a few minor suggestions.

Comment 1:  Abstract, the reporting of the logistic regression results can be more informative by specifying what factors are associated with the outcome variables. Similarly, the results of the chi-square test can also be clarified.

Response: This has been amended accordingly as suggested by the reviewer. Kindly refer to the abstract section, lines 43-48, page 2.

Comment 2: The authors did not mention anything about "rural primary healthcare clinics" in the results part, however, they suddenly raised this issue in the discussion part, which is weird and should be revised.

Response: This has been amended accordingly. Kindly refer to results section, lines 226-230, page 9.

Comment 3: Introduction: third paragraph, can the authors give more explanation on why the use of mHealth applications can reduce the spread of SARS-CoV-2 and other infectious diseases?

Response:  Further explanation has been provided accordingly as suggested by the reviewer. Kindly refer to introduction section: third paragraph, lines 80-86, page 3.

Comment 4: Figures 2-5 are hard to read, the fonts are too small.

Response: The fonts of the figures have been amended. Kindly refer to pages 10, 12, 17 & 24

Comment 5: There are many formatting problems with the references

Response: The reference has been formatted according to the journal’s referencing style.
